# Extraction of Union and Intersection Axioms from Biomedical Text

Nikhil Sachdeva ✉[0000−0001−7544−4552], Monika Jain[0000−0001−7697−7772], and Raghava Mutharaju[0000−0003−2421−3935]

Knowledgeable Computing and Reasoning Lab, IIIT-Delhi, India.
{nikhil16061, monikaja, raghava.mutharaju}@iiitd.ac.in

**Abstract.** Many ontology, especially the ones created automatically by the ontology learning systems, have only shallow relationships between the concepts, i.e., simple subclass relations. Expressive axioms such as the class union and intersection are not part of the ontology. These expressive axioms make the ontology rich and play an essential role in the performance of downstream applications. However, such relations can generally be found in the text documents. We propose a mechanism and discuss our initial results in extracting union and intersection axioms from biomedical text using entity linking and taxonomic tree search.

**Keywords:** Ontology Learning · Ontology Enrichment · Axiom Extraction.

## 1 Introduction

*Ontology learning* [4] is the process of building ontologies automatically from text. Several ontology learning systems [5,1] such as Text2Onto[1], Doodle OWL[2] and DL-Learner[3] have been developed. Most of these systems support learning of classes, subclasses and taxonomic relationships. But they do not support mining of more expressive axioms from text such as union, intersection, quantifiers and cardinality relation among the concepts. A richer and more expressive ontology can be very useful to the downstream applications such as recommendation systems and question and answering systems.

We propose a mechanism to extract union and intersection axioms from the text with the help of an ontology. The extracted axioms are then added to the ontology to enhance its expressivity. Examples of intersection and union axioms are given in Axioms 1 and 2. Axiom 1 models the information that a *Mixed Glioma* (type of tumor) is a combination of *Astrocytoma* and *Oligodendroglioma*. Axiom 2 captures the information that a *Tumor* can be either *Benign* or *Pre-Malignant* or *Malignant*.

---

[1] http://neon-toolkit.org/wiki/1.x/Text2Onto.html
[2] https://sourceforge.net/projects/doddle-owl/
[3] http://dl-learner.org/

$$Mixed\ Glioma \sqsubseteq Astrocytoma \sqcap Oligodendroglioma \tag{1}$$

$$Tumor \sqsubseteq Benign \sqcup PreMalignant \sqcup Malignant \tag{2}$$

Although there have been attempts at identifying concepts and relations in the text with the help of an ontology [3] and word embeddings [6], to the best of our knowledge, this is the first attempt at extracting complex (non-taxonomic) ontology axioms from the text. In the next section, we describe the axiom extraction pipeline followed by the discussion of results.

## 2  Approach

Our system expects an ontology along with one or more text documents relevant to the ontology as input. The system identifies the named entities in the text that are of interest (based on the concepts defined in the ontology) and checks for potential union and intersection axioms. If an appropriate match is found, our system generates the axioms as shown in Figure 1. The first step is to extract the named entities from the text and perform entity linking, wherein each entity extracted from the text is assigned a type (a concept in the ontology). The entities were extracted using the SpaCy models for biomedical text processing[4] and the Metamap application[5] is used for recognizing those entities in the UMLS Metathesaurus[6]. Metathesaurus is an semantic network of biomedical entities taken from more than 200 vocabularies. It provides various definitions, taxonomic and non-taxonomic relations for every entity present in the network.

Some of the extracted entities from the text may not be associated with any of the concepts in the ontology. Entity mentions in the text are represented by $E = \{e_1, e_2, \ldots, e_n\}$ and the concepts in the ontology are represented by $C = \{c_1, c_2, \ldots, c_n\}$. All the concepts are considered as potential candidates for the entities to be linked. The pseudocode for entity linking is given in Algorithm 1.

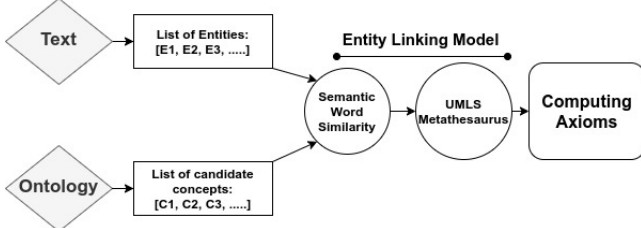

Fig. 1: Architecture for extracting union and intersection axioms from the text

---

[4] https://allenai.github.io/scispacy/

[5] https://metamap.nlm.nih.gov/

[6] https://www.nlm.nih.gov/research/umls/index.html

---

**Algorithm 1** Linking the entity mentions in the text to the concepts in the ontology

---

1: **for** $e_i \in entities = e_1, e_2, \ldots, e_n$ **do**
2:     **for** $c_j \in concepts = c_1, c_2, \ldots, c_m$ **do**
3:         $score_B \leftarrow cos_B(e_i, c_j)$                         ▷ BioWordVec Model
4:         $score_C \leftarrow cos_C(e_i, c_j)$                  ▷ Custom Word2Vec Model
5:         **if** $score_B \geq \alpha$ **AND** $score_C \geq \beta$ **then**
6:             **if** $e_i$ **isDescendantOf** $c_j \in$ **UMLS then**
7:                 $addPairToOntology(e_i, c_j)$
8:             **else**                ▷ $\alpha, \beta, \gamma$ are adjustable threshold parameters
9:                 $L \leftarrow lowestCommonAncestor(e_i, c_j)$
10:                $M_e \leftarrow Metamap(e_i, L)$
11:                $M_c \leftarrow Metamap(c_j, L)$
12:                **if** $(M_e + M_c)/2 \geq \gamma$ **then**
13:                     $addPairToOntology(e_i, c_j)$

---

Entity linking consists of three stages. Each pair of entity mention and concept $(e_i, c_i)$ will go through all the three stages to determine if the entity mention $e_i$ is an instance of the candidate concept $c_i$. In the first stage, the cosine distance of real-valued word embeddings of $e_i$ are compared with $c_i$ to determine if they have a qualitative semantic similarity. The comparison is made using the embeddings generated from BioWordVec[7] and a custom Word2Vec[8] model. The former model captures the contextual information around an entity from the unlabelled biomedical text using the MeSH vocabulary. The custom Word2Vec model is trained over only those biomedical text articles that contain the concepts in the given ontology. If the pair $(e_i, c_i)$ satisfies the minimum threshold values $(\alpha, \beta$ from Algorithm 1), it moves to the next stage where we check whether $e_i$ is an instance of $c_i$ using UMLS Metathesaurus and Metamap (Figures 2a and 2b). In *scenario-1*, if $c_i$ is present in the UMLS tree, we check if $e_i$ matches any of the descendants of $c_i$ in the tree. Figure 2a shows that entity *Congenital Bacterial Pneumonia* is a descendant of the concept *Pneumonia*. In *scenario-2*, where $e_i$ is not a direct descendant of $c_i$, we find the lowest common ancestor $l_i$ of $e_i$ and $c_i$. Figure 2b shows that the entity *Basal Pneumonia* and the concept *Infective Pneumonia* have a common ancestor *Respiratory Finding*. Using Metamap, we compare the number of semantic groups[9] the pairs $(e_i, l_i)$ and $(c_i, l_i)$ share. A semantic group is a broader group that a term (entity or concept in our case) can be a part of, according to Metamap. Further, we compare the UMLS generated context vectors of $e_i, c_i$ and $l_i$ (taken two at a time), using cosine measure. Based on these scores, we determine if $e_i$ is an instance of $c_i$. The hyperparameter $\gamma$ represents the neutralized score of these comparisons. The value of this hyperparameter is adjustable and can be set on the basis of experiments.

---

[7] https://github.com/ncbi-nlp/BioWordVec
[8] https://code.google.com/archive/p/word2vec/
[9] https://metamap.nlm.nih.gov/SemanticTypesAndGroups.shtml

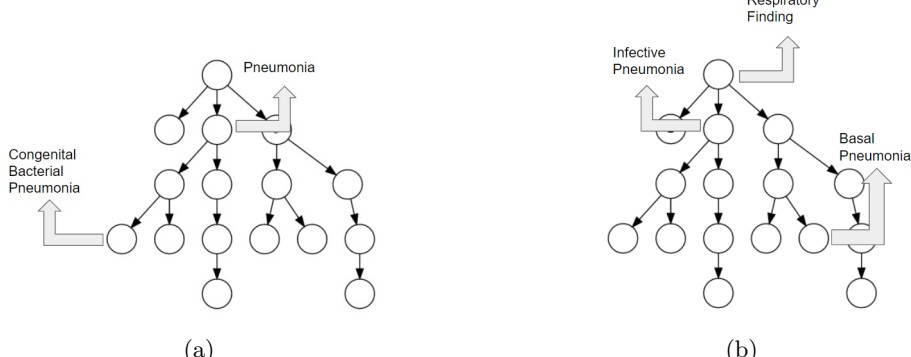

(a)                                              (b)

Fig. 2: (a) represents scenario-1 where entity $e$ is a descendant of concept $c$ and (b) represents scenario-2 where both Entity $e$ and Concept $c$ have a common ancestor $L$.

Finally, when a pair $(e_i, c_i)$ is added in the ontology, $e_i$ is also added as an instance of all the parent concepts of $c_i$. After repeating these three stages for each pair of $e_i$ and $c_i$, we get a list of concepts with their corresponding sets of instances (entities). Using these instances, we compute union and intersection of the concepts based on the below formula

$$total\_sets = \binom{n}{1} + \binom{n}{2} + \binom{n}{3} + \binom{n}{4} + \binom{n}{5} + \binom{n}{6} \tag{3}$$

where each $\binom{n}{i}$ represents sets of $i$ concepts. Each of these sets is compared with every candidate concept to check for union and intersection axioms. For example, to obtain Axiom 1, the intersection of instances of *Astrocytoma* and *Oligodendroglioma* is compared with the instances of *Mixed Glioma*. If the latter is a subset of the former, we can add this axiom to the ontology. We have considered such combinations up to a size of 6. The size was determined using experiments by comparing the F1 scores of sets of different size.

## 3   Results and Discussion

We could not find any biomedical ontology that has union and intersection axioms along with the concepts that have instances for evaluating our model. Moreover, there is no dataset having such ontologies and the corresponding text corpora in the medical domain. We choose the Disease ontology[10] as it is rich in such axioms, but it lacks the concept-instance pairs. So we made an approximation here and preprocessed the ontology. We observed that the lowermost leaf-child

---

[10] https://disease-ontology.org/

concept could act as an instance of the directly connected parent class. Subsequently, it can be connected as an instance of all the subsequent parent classes in the hierarchy. This process is executed for all the leaf-nodes in the ontology. We extracted 739 articles from PubMed Central[11] based on the concepts in the Disease Ontology. Within the ontology, there are a total of 10,085 intersection axioms and 323 union axioms. Based on the extracted articles and the Disease ontology, the F1 scores for the union axioms is 0.142, and for the intersection axioms, it is 0.1908.

While analyzing the results, we observed that many false positive axioms were generated, hence reducing the model's precision. One of the reasons is that many trivial union axioms were generated that contained many general classes and this increased the number of false positive axioms. Furthermore, there is the unavailability of a proper dataset consisting of a complete and rich ontology with the corresponding text corpora. Therefore, constructing a relevant dataset and defining baselines are some of the tasks of our future work. Furthermore, to improve the architecture, we are working on incorporating English language pattern heuristics where we can apply the syntactical rules to extract axioms from unstructured text robustly. We are also working on fine-tuned deep learning models to improve the contextual word embeddings for the target word. Based on our results, we plan to apply neural models like BERT [2] that use transformers to generate a language model with strong contextual relations between the words of the text.

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
