# OpenReview forum: "Extraction of Union and Intersection Axioms from Biomedical Text"
_eswc-conferences.org/ESWC/2021/Conference/Poster_and_Demo_Track — ESWC2021 P&D_

### Official Review · AnonReviewer1 · 2021-04-06
**A bit overclaimed**

**Rating:** 4
**Confidence:** 3

**Review:**

The paper extracts union and intersection axioms from text. A heuristic entity linking algorithm is presented. Union and intersection axioms are found by set operations over extracted entities.

The following issues should be clarified.

(1) Some claims in the paper may be a bit over. For example, the authors said "we could not find any ontology that has union and intersection axioms along with the concepts that have instances", while the well-known Wine ontology in the OWL guide is clearly one such ontology. Another example: "this is the first attempt at extracting complex ontology axioms from the text." It would depend on how to define "complex".

(2) Some key steps in the present algorithms are not clear. What is Metamap in Algorithm 1? How do you "determine if e_i js an instance of c_i" based on their lowest common ancestor?

(3) According to Equation (3), many trivial union axioms may be generated. It would be good to have a discussion here.

(4) For the experiments the authors could not find "a proper dataset consisting of a complete and rich ontology with the corresponding text corpora." Would it challenge the practicability of the proposed approach?

**Anonymity:**

Yes, I would like my review to remain anonymous.

---

### Official Review · ~Edelweis_Rohrer1 · 2021-04-13
**The presented approach appears as useful to make an existing ontology more meaningful, but is is not clearly introduced.**

**Rating:** 6
**Confidence:** 3

**Review:**

Summary

The paper proposes a mechanism to extract union and intersection axioms from a text, looking for concepts of a reference ontology. As a result, an ontology is enriched with the extracted axioms. As most of ontology learning systems have the capability of extracting classes and the hierarchical relations among classes, the motivation of the proposed approach is to extract more expressive relations.

Evaluation

Both in the Introduction and in the beginning of Section 2, the proposed approach is not clearly introduced due to the following:
-	It is not introduced that two ontologies that play different roles participate in the process: the ontology that is going to be enriched and the reference ontology UMLS Metathesaurus.
-	The role that plays Metamap is not properly introduced neither.

However, after presenting the pseudocode of the algorithm, the idea behind the proposed approach is quite clear.
Finally, the algorithm could not be applied as it is thought because of the lack of ontologies with a corresponding text corpora in the medical domain. Anyway some results were obtained with the approximation made considering leaf-child concepts of an ontology as they were instances.

Final comments

The presented approach appears as useful to make an existing ontology more meaningful. However, the approach is not clearly introduced; the role of basic elements that participate in the process is deduced from the description of the algorithm.


**Anonymity:**

No, I would like my review to be deanonymized.

---

### Official Review · AnonReviewer3 · 2021-04-16
**Good paper with interesting approach to ontology building with extracted biomedical text.**

**Rating:** 7
**Confidence:** 5

**Review:**

The paper discusses an approach to extract simple axioms from text and add them to a biomedical ontology. While the paper offers some interesting aspects and insights into axiom extraction from biomedical text, it could be improved with the following additions:

- abstract is missing
- the paper claims that no text mining is done with mapping to ontological concepts, but fails to mention LexMappr, or papers such as https://thesai.org/Downloads/Volume3No12/Paper_6-Simple_Method_for_Ontology_Automatic_Extraction_from_Documents.pdf
- it is not clear how the entities are extracted from text
- add links to BioWordVec and Word2Vec
- explain minimum threshold values alpha and beta
- explain why an entity has to have cosine similarity with both Word2Vec and BioWordVec
- a link to the experiments that determined the size of combinations that are compared to find the axiom relation

Overall good paper, that could benefit from a bit of restructuring and some additional links.

**Anonymity:**

Yes, I would like my review to remain anonymous.

---

### Decision · Program_Chairs · 2021-04-19

Accept